# A vibrissa pathway that activates the limbic system

**Michaël Elbaz[1†], Amalia Callado Perez[1,2†], Maxime Demers[1], Shengli Zhao[3], Conrad Foo[2], David Kleinfeld[2,4]\*, Martin Deschenes[1]\***

[1]CERVO Research Center, Laval University, Québec City, Canada; [2]Department of Physics, University of California, San Diego, San Diego, United States; [3]Department of Neurobiology, Duke University Medical Center, Durham, United States; [4]Section of Neurobiology, University of California, San Diego, San Diego, United States

**Abstract** Vibrissa sensory inputs play a central role in driving rodent behavior. These inputs transit through the sensory trigeminal nuclei, which give rise to the ascending lemniscal and paralemniscal pathways. While lemniscal projections are somatotopically mapped from brainstem to cortex, those of the paralemniscal pathway are more widely distributed. Yet the extent and topography of paralemniscal projections are unknown, along with the potential role of these projections in controlling behavior. Here, we used viral tracers to map paralemniscal projections. We find that this pathway broadcasts vibrissa-based sensory signals to brainstem regions that are involved in the regulation of autonomic functions and to forebrain regions that are involved in the expression of emotional reactions. We further provide evidence that GABAergic cells of the Kölliker-Fuse nucleus gate trigeminal sensory input in the paralemniscal pathway via a mechanism of presynaptic or extra-synaptic inhibition.

**\*For correspondence:**
dk@physics.ucsd.edu (DK);
martin.deschenes@fmed.ulaval.ca (MD)

[†]These authors contributed equally to this work

**Competing interest:** The authors declare that no competing interests exist.

## Editor's evaluation

Elbaz and colleagues show that the interpolaris subdivision of the trigeminal brainstem innervates not only somatosensory thalamus but also other midbrain and hindbrain structures. A key target of interpolaris appears anatomically to be excitatory projection neurons of the Kolliker-Fuse nucleus. Physiological recordings from the dorsal medullary reticular nucleus, Kolliker-Fuse nucleus, and the central amygdala reveal responses to vibrissa deflection, suggesting a role for this pathway in limbic system behaviors.

## Introduction

Most sensory systems comprise parallel pathways of sensory information that encode different features of a stimulus and also take part in the control of sensor motion (*Merigan and Maunsell, 1993*; *Lomber and Malhotra, 2008*; *Nassi and Callaway, 2009*; *Niu et al., 2013*; *Igarashi et al., 2012*). The vibrissa system of rodents is no exception (reviewed in *Kleinfeld and Deschênes, 2011*). Ascending signals in the vibrissa system travel along two main trigeminothalamic pathways (reviewed in *Prescott et al., 2016*): (1) a lemniscal pathway that arises from the trigeminal nucleus principalis (PrV), transits through the ventral posterior medial nucleus (VPM) of the thalamus, and projects to the primary somatosensory cortex; (2) a paralemniscal pathway that arises from the rostral part of trigeminal nucleus interpolaris (SpVIr), transits through the posterior group (Po) of the thalamus, and projects to the somatosensory cortical areas and to the vibrissa motor cortex.

In contrast with PrV cells, which innervate principally VPM thalamus, SpVIr cells that project to Po thalamus also innervate a number of additional regions by means of branching axons (*Pierret et al.,*

*2000*; *Veinante et al., 2000*; *Bellavance et al., 2017*). These include the superior colliculus, the anterior pretectal nucleus, the ventral division of zona incerta, and the dorsal lateral sector of the facial nucleus. Tract-tracing studies by means of classic tracers also reported interpolaris projections to the brainstem and spinal cord (*Jacquin et al., 1986*; *Phelan and Falls, 1991*). Whether the latter projections also arise from Po-projecting cells remains an open question.

While the lemniscal pathway conveys tactile information, as well as information about the relative phase of the vibrissae in the whisk cycle (*Yu et al., 2006*; *Curtis and Kleinfeld, 2009*; *Khatri et al., 2010*; *Moore et al., 2015*; *Isett and Feldman, 2020*), the role of the paralemniscal pathway remains puzzling. It was proposed that this pathway conveys information about whisking kinematics (*Yu et al., 2006*; *Golomb et al., 2006*), but later studies found that encoding of whisking along the paralemniscal pathway is relatively poor (*Moore et al., 2015*; *Urbain et al., 2015*). It was also proposed that the paralemniscal pathway is specifically activated upon noxious stimulation (*Masri et al., 2009*; *Frangeul et al., 2014*), but it has never been shown that interpolaris cells that respond to vibrissa deflection are also activated by noxious stimuli. Thus, the general function of the paralemnical pathway remains unresolved.

Here, we used virus-based tract-tracing methods and electrophysiology to document the full extent of the collateral projections of interpolaris cells that give rise to the paralemniscal pathway. We searched for new pathways of ascending information to forebrain regions, with a focus on pathways that include the Kölliker-Fuse/parabrachial complex (KF/PBc). The KF/PBc were previously described primarily in terms of their role in eliciting changes in respiration and cardiac function (reviewed in *Saper and Stornetta, 2015*). Yet the role of inhalation in either driving or pacing whisking (*Deschênes et al., 2012*; *Moore et al., 2013*) provides motivation to investigate if the KF/PBc has a broader role in orofacial motor actions. Further motivation comes from the projections of the KF nucleus to the trigeminal sensory nuclei (*Geerling et al., 2017*), which contains multiple pathways for the control of motoneurons involved in whisking (*Bellavance et al., 2017*). Finally, we searched for feedback projections from targets along the paralemnical pathways to brainstem nuclei as a means to uncover a potential role of this pathway in rat's behavior.

## Results

### Collateral projections of interpolaris cells that project to Po thalamus

To map the collateral projections of interpolaris cells that innervate Po thalamus, we injected the retrograde virus G-pseudotyped Lenti-Cre in Po thalamus, and AAV2/8-hSyn-DIO-GFP in the vibrissa-responsive sector of subnucleus SpVIr (five juvenile rats; *Figure 1A*). This approach revealed an unexpected diversity of axonal projections across all animals (*Figure 1B–G*). In accordance with prior studies, terminal fields are found in Po thalamus, the ventral division of zona incerta, the anterior pretectal nucleus, the superior colliculus, the perirubral region, the dorsal lateral part of the facial nucleus, and in the other subdivisions of the sensory trigeminal complex (*Jacquin et al., 1986*; *Jacquin et al., 1989*; *Veinante et al., 2000*; *Bellavance et al., 2017*). As new results, profuse projections are also found in the ipsilateral dorsal medullary reticular (MdD) nucleus and further caudally in the dorsal horn of the cervical cord. An ascending projection terminates ipsilaterally in the intertrigeminal region (ITr) and in the KF/PBc. Whether individual interpolaris cells project to many or all of the above-mentioned target regions remains an open issue. What is clear, however, is that interpolaris cells that project to Po thalamus broadcast sensory messages to multiple midbrain and hindbrain regions.

Injection of the virus AAV1-hSyn-eGFP-WPRE in the vibrissa-responsive sector of subnucleus SpVIr (three adult rats; *Figure 1—figure supplement 1*) yielded similar results to those above (*Figure 1B, G*). Yet anterograde labeling is now found in the cerebellum. This is consistent with prior studies, which reported that different populations of SpVI cells project to the thalamus and cerebellum (*Steindler, 1985*; *Jacquin et al., 1986*). Of note, in sagittal sections of the brainstem, the terminal field of interpolaris axons in the KF/PBc covers an extensive crescent-shaped territory at the rostral border of the PrV and trigeminal motor nuclei (*Figure 1—figure supplement 1*).

### KF/PBc and MdD cells respond to vibrissa deflection

Interpolaris cells that give rise to the paralemniscal pathway are known to respond to vibrissa deflection (*Veinante et al., 2000*). Thus, one expects KF/PBc and MdD neurons to also respond to vibrissa

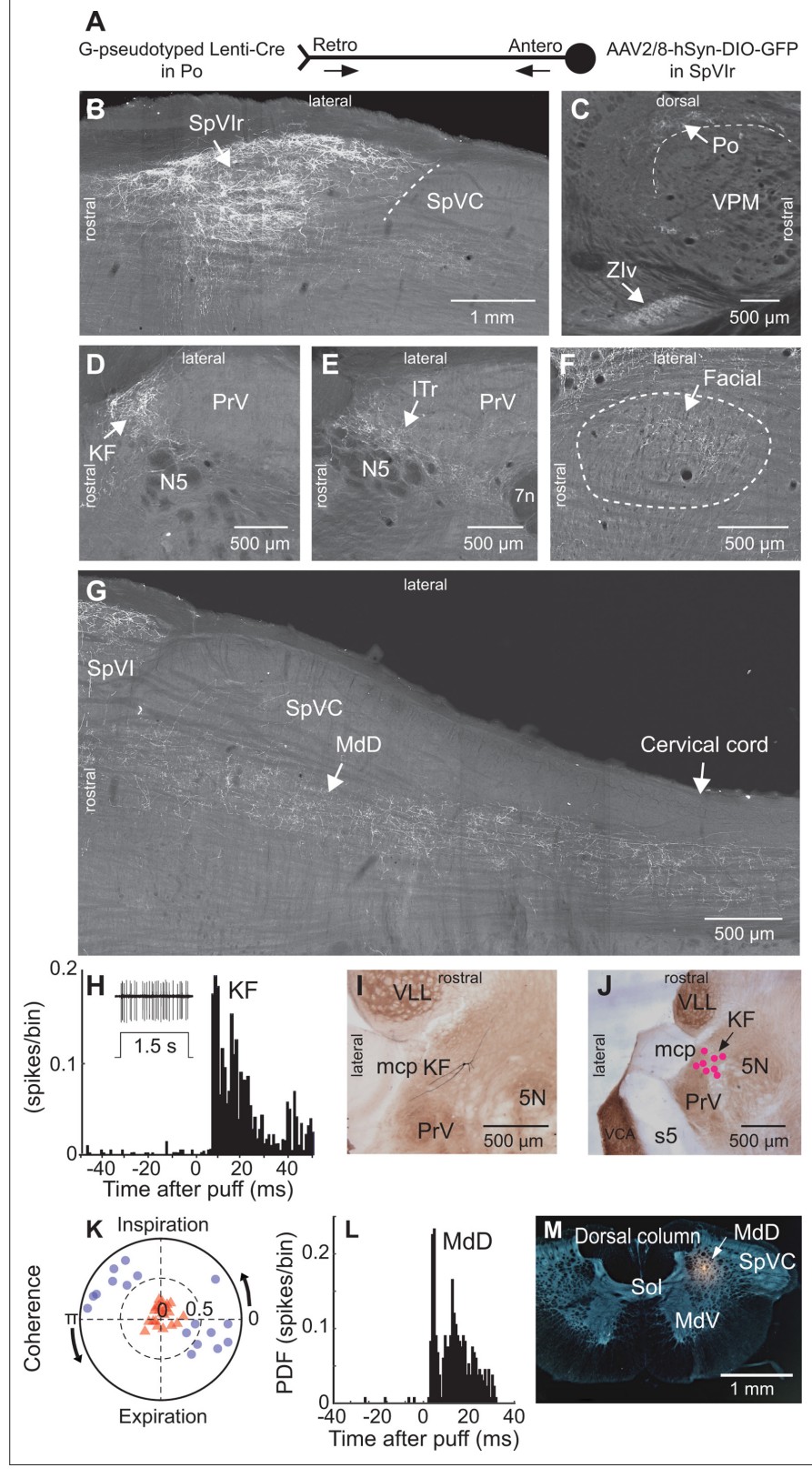

**Figure 1.** Anatomical and electrophysiological evidence that vibrissa-responsive interpolaris cells have widespread axonal projections. (**A**) Viral method used for labeling paralemniscal projections. (**B**) Labeling of interpolaris cells after injection of G-pseudotyped Lenti-Cre virus in Po thalamus, and a Cre-dependent AAV that expresses GFP in the vibrissa-responsive sector of SpVIr. Horizontal section. (**C**) Anterograde labeling of terminal fields

*Figure 1 continued on next page*

*Figure 1 continued*

in Po thalamus and zona incerta. Sagittal section. (**D**) Anterograde labeling in the KF/PBc. Horizontal section. (**E**) Anterograde labeling in the ITr. Horizontal section. (**F**) Anterograde labeling in the dorsal sector of the facial nucleus. Horizontal section. (**G**) Anterograde labeling in the MdD and cervical cord. Horizontal section. (**H**) Population peristimulus time histogram of spike discharges evoked in KF (22 cells) by air puff deflection of the vibrissae in the anesthetized rat. A representative response is shown in the insert. (**I**) Example of a vibrissa responsive KF cell labeled by juxtacellular delivery of Neurobiotin. Horizontal section. (**J**) Location of eight juxtacellularly labeled KF cells. Horizontal brainstem sections in (**I**) and (**J**) were counterstained for cytochrome oxidase. Horizontal section. (**K**) Spectral coherence of spontaneous discharges of KF cells with the respiratory cycle at the respiratory frequency; 1–3 Hz. Note that, in contrast with the respiratory units (blue dots), spontaneous discharges of vibrissa-responsive cells (red triangles) display low coherence with respiration. (**L**) Population peristimulus time histogram of spike discharges evoked in MdD (33 cells) by air puff deflection of the vibrissae. (**M**) Recording site in the MdD labeled by an iontophoretic injection of Chicago Sky Blue. This coronal section was counterstained for cytochrome oxidase and a negative image was generated. Coronal section. See *Figure 1— figure supplement 1* for additional anatomical data. Abbreviations for all anatomy: 5n, root of the trigeminal motor nucleus; 5N, trigeminal motor nucleus; 5t, trigeminal tract; 7n, facial nerve tract; 7N, facial nucleus; Amb, ambiguus nucleus; APT, anterior pretectal nucleus; BSTL, bed nucleus of the stria terminalis; CeA, central amygdala; Cerv Cord, cervical cord; CM/PC, central medial/paracentral thalamic nuclei; CPu, caudate putamen; DR, dorsal raphe; EW, Edinger-Westphal; Hab, habenula; IML, intermedio-lateral column of the spinal cord; ITr, intertrigeminal region; KF, Kölliker-Fuse nucleus; KF/PBc, Kölliker-Fuse/parabrachial complex; mcp, middle cerebellar peduncle; MdD, dorsal part of the medullary reticular formation; MdV, ventral part of the medullary reticular formation; mt, mammillothalamic tract; NA, nucleus ammbiguus; NTS, nucleus of the solitary tract; opt, optic tract; PAG, periaqueductal gray; PB, parabrachial nuclei; PC, paracentral thalamic nucleus; PCRt, parvicellular reticular formation; PLH, posterior lateral hypothalamus; Po, posterior nuclear group of the thalamus; PrV, principal trigeminal nucleus; RN, red nucleus; Rt, reticular thalamic nucleus; s5, sensory root of the trigeminal nerve; SC, superior colliculus; scp, superior cerebellar peduncle; SpVC, caudalis division of the spinal trigeminal complex; Sol, nucleus of the solitary tract; SpVIc, caudal sector of the interpolaris trigeminal nucleus; SpVIr, rostral division of the interpolaris nucleus; TG, trigeminal ganglion; VLL, ventral nucleus of the lateral lemniscus; VPL, ventral posterolateral thalamic nucleus; VPM, ventral posterior medial nucleus; VPCc, parvicellular sector of the ventral posteromedial thalamic nucleus; VPPc, parvocellular part of the ventral posterior thalamic nucleus; VRG, ventral respiratory group; vsc, ventral spinocerebellar tract; ZIv, ventral division of zona incerta.

The online version of this article includes the following figure supplement(s) for figure 1:

**Figure supplement 1.** Interpolaris cells project to the KF/PBc and MdD (Supplementary Information related to *Figure 1*).

deflection. We indeed found vibrissa-responsive neurons in the KF/PBc (latency: 8.0±1.2 ms; mean ± SD; 22 cells across 14 adult rats; *Figure 1H* K) and in the MdD nucleus (latency: 6.0±1.4 ms; 33 cells across three additional adult rats; *Figure 1L and M*) of ketamine/xylazine anesthetized animals. The location of recorded units was assessed by single-cell juxtacellular labeling in the KF/PBc (eight cells in 8 of the 14 rats), and by iontophoretic deposit of Chicago Sky Blue in the MdD nucleus (three rats). Like interpolaris cells, the receptive field of KF/PBc and MdD cells includes multiple vibrissae. In the KF/PBc, vibrissa-responsive cells are intermingled with cells that discharge in phase with the respiratory rhythm. Yet the spontaneous activity of vibrissa-responsive cells displays no statistically significant spectral coherence with respiration (mean coherence< C > in the spectral band of breathing is |< C >|=0.16 for vibrissa-responsive cells versus |< C >|=0.72 for respiratory units (*Figure 1K*)).

We did not investigate the parameters of vibrissa deflection that best drive KF/PBc and MdD cells. Our goal was to cross-check the tract-tracing results (*Figure 1D*), which indicate that some KF/PBc and MdD cells receive input from vibrissa-responsive interpolaris neurons.

## Axonal projections of KF/PBc cells that receive interpolaris input s

Among the targets of the paralemniscal pathway, we focused on the KF/PBc because of the well-documented hodology of its axonal projections (see review by *Saper and Stornetta, 2015*). As a means to visualize the axonal projections of KF/PBc cells that receive interpolaris input, we injected AAV2/1 that expresses Cre in the vibrissa-responsive sector of subnucleus SpVI to achieve antero-grade transsynaptic expression of Cre (*Zingg et al., 2017*), and AAV1-EF1a-DIO-hChR2-eYFP was injected in the KF/PBc (three adult rats; transsynaptically labeled cells amount to 55, 62, and 68 in each of the rats). We observed transsynaptic anterograde labeling in the lateral part of the facial

nucleus, throughout the ventral lateral medulla, the MdD nucleus, the nucleus of the solitary tract, and the cervical cord in all animals (*Figure 2A–D*). Ascending projections were found in the dorsal raphe, the Edinger-Wesphal nucleus, the periaqueductal gray, the posterior lateral hypothalamus, the paracentral and central medial thalamic nuclei, the parvicellular part of the ventral posterior thalamic nucleus, the ventral division of zona incerta, the central amygdala (CeA), and the lateral part of the bed nucleus of stria terminalis in all animals (*Figure 2E–G - Figure 2—figure supplement 1*). In summary, KF/PBc cells that receive interpolaris input in turn project to most of the brainstem and fore-brain regions previously identified by means of anterograde tracer injections in the KF/PBc (*Saper and Stornetta, 2015*; *Figure 2H*). Yet, it is worth noting that no anterograde labeling is observed in the trigeminal sensory nuclei. This negative result is of critical importance. It indicates that KF/PBc cells that receive interpolaris input do not feedback to the relay cells that give rise to the paralemniscal pathway (see below).

As PB cells that project to the CeA receive input from vibrissa-responsive SpVI neurons, one expects to find vibrissa-evoked responses in the amygdala. We addressed this issue by recording the local field potential evoked by air jet deflection of the vibrissae (three adult rats; *Figure 3A* C). These experiments were carried out under urethane instead of ketamine/xylazine anesthesia as $\alpha_2$ agonists, for example, xylazine, strongly depress synaptic transmission in the parabrachial-amygdaloid pathway (*Delaney et al., 2007*). Vibrissa deflection evoked a clear negative field potential in the CeA (peak latency of 12 ms as an average over all three rats and 50 trials per rat; *Figure 3A and B*). This response was abolished in all animals following electrolytic lesion of the ipsilateral PB nucleus (*Figure 3A and C*). Taken together, our anatomical (*Figure 2*) and electrophysiological (*Figure 3A–C*) data delineate a pathway of vibrissa information that reaches the CeA and several limbic regions of the forebrain via a relay in the KF/PBc.

## Differential projections of KF and PB cells

Prior studies in mice have shown that separate pools of KF/PBc neurons project to the forebrain and to the lower brainstem (*Geerling et al., 2017*; *Barik et al., 2018*). We thus examined whether descending and ascending projections of the KF/PBc in rats arise from different or overlapping neurons. We made large injections, i.e., 100 nl, of the retrograde labels retroAAV-CAG-eGFP and retroAAV-CAG-mCherry in the CeA and the facial nucleus, respectively (two rats; *Figure 3D–G*). Although a number of retrogradely labeled cells were found in the KF/PBc, none was doubly labeled. In detail, amygdala-projecting cells (326 in rat 1 and 312 in rat 2) were concentrated in lateral and medial PB nucleus, while the majority of facial projecting cells (299 in rat 1 and 140 in rat 2) were located in KF nucleus. In complementary experiments, we injected retroAAV expressing Cre in the ventral respiratory group and AAV2/8-hSyn-DIO-eGFP in the KF/PBc (two rats). Anterograde labeling was observed in the facial nucleus, the ventral respiratory group, and the MdD nucleus (*Figure 3—figure supplement 1*), but not in midbrain and forebrain regions. Taken together, these results confirm that neighboring but separate populations of KF/PBc neurons project to the forebrain and to the lower brainstem (*Figure 3H*).

## GABAergic KF cells gate sensory transmission in subnucleus SpVI

In rodents, KF projections to the brainstem arise from two neuronal populations: glutamatergic cells that project to the autonomic, respiratory, and motor regions of the medulla, and GABAergic neurons, which project principally to subnucleus SpVI (*Geerling et al., 2017*). To determine whether GABAergic KF cells inhibit interpolaris cells, we made large injections, i.e., 200 nl, of the transsyn-aptic label anteroAAV-expressing Cre (*Zingg et al., 2017*) in the KF/PBc and AAV2-Ef1a-DIO-eYFP in subnucleus SpVI (three rats; *Figure 4A–D*). Unexpectedly, few interpolaris neurons were labeled (44, 48, and 53 cells, respectively, in each of three rats). Anterograde labeling was observed in the KF/PBc and in the parvicellular part of the ventral posterior thalamic nucleus, a region known to process input from the tongue and oral cavity (*Ogawa and Nomura, 1988*; *Verhagen et al., 2003*), in all animals. No labeling was observed in the cerebellum, nor in brainstem and midbrain regions that receive input from the paralemniscal pathway (*Figure 2H*). Yet, cells in the parvicellular reticular formation (PCRt) at the medial border of the trigeminal nuclei were extensively labeled, which indicates that the paucity of cell labeling in subnucleus SpVI is hardly ascribable to a technical pitfall. Taken together, these results indicate that KF projections do not innervate the SpVI interneurons or the projection cells that give rise to the paralemniscal pathway.

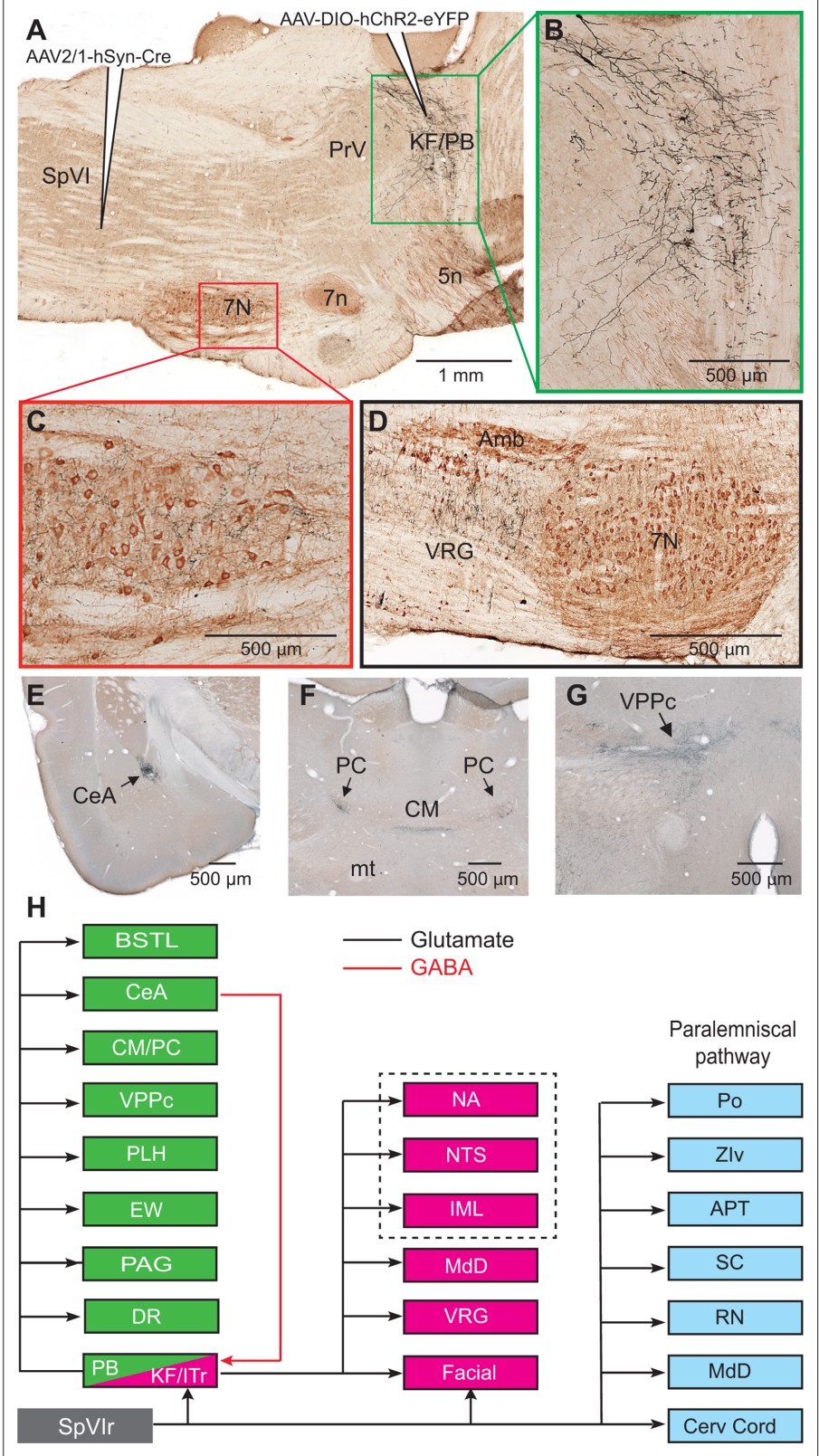

**Figure 2.** Axonal projections of KF/PBc cells that receive interpolaris input. See *Figure 1* for abbreviations. (**A–C**) AAV2/1-hSyn-Cre-WPRE was injected in the vibrissa-responsive sector of the SpVIr, and AAV2/1-EF1a-DIO-hChR2-eYFP was injected in the KF/PBc. This parasagittal section shows labeling in the KF/PBc. The section was immunostained for choline acetyltransferase (**A**). The green and red framed areas are enlarged in (**B**) and

*Figure 2 continued on next page*

*Figure 2 continued*

(**C**). Sagittal section. (**D**) Terminal labeling in the ventral respiratory group underneath the ambiguous nucleus. Sagittal section. (**E**) Anterograde labeling in the central amygdala. Coronal section. (**F**) Anterograde labeling in the paracentral and central medial thalamic nuclei. Coronal section. (**G**) Anterograde labeling in the parvocellular division of the ventral posterior medial nucleus of the thalamus (see Figure S2 for additional projection sites). Coronal section. (**H**) Summary of the first-order axonal projections of interpolaris cells and the second-order projections that derive from the KF/PBc. Projection sites in the framed area are from prior studies (see review by *Saper and Stornetta, 2015*). The classical paralemniscal pathway, that is, structures that receive input from SpVI, is in cyan. Magenta is the brainstem target of KF and green is the target of PG. MdD appears twice as it received input from both SpVI and the KF/PBc. We have added this to the legend.

The online version of this article includes the following figure supplement(s) for figure 2:

**Figure supplement 1.** Additional projection sites of KF/PBc cells that receive interpolaris input (Supplementary Information related to *Figure 2*).

To further assess these anatomical results, we injected AAV-hSyn-ChR2-eYFP in the KF nucleus and used an optrode to concurrently stimulate KF axons and record the activity of SpVI neurons (three adult rats; *Figure 4E–G*). Interpolaris cells either discharged spontaneously or were driven by jiggling a single vibrissa with a piezoelectric stimulator. Optogenetic stimulation suppressed sensory-evoked discharges in monovibrissa-responsive cells (26 cells in the same three rats, t-test yields p<0.001; lower panel in *Figure 4G*) but did not affect the firing of multivibrissa-responsive units (32 cells in the same three rats, p=0.84; *Figure 4G*). Together with the lack of transsynaptic labeling of local circuit cells from the KF nucleus, these results raise the possibility that KF-induced suppression of sensory responses in local circuit interpolaris cells is mediated by presynaptic or extrasynaptic inhibition, as summarized in *Figure 4H*.

## Discussion

Our study reveals a hitherto unsuspected diversity of axonal projections of interpolaris cells that give rise to the paralemniscal pathway. We identified two new collateral projections that target the MdD nucleus and the KF/PBc (*Figures 1 and 2*). The diagram in *Figure 2H* summarizes the first-order axonal projections of interpolaris cells that give rise to the paralemniscal pathway, and the second-order projections that derive from the KF/PBc. Furthermore, we show that the KF/PBc contains two populations of vibrissa-responsive neurons. One population gives rise to an ascending pathway, which projects to limbic regions of the forebrain, and the other one projects to the autonomic, respiratory, and motor regions of the medulla (*Figure 3*); the diagram in *Figure 3H* summarizes this circuit. Finally, the KF nucleus contains a pool of GABAergic neurons that do not receive interpolaris input, but project locally and to the trigeminal sensory nuclei (*Figure 4*); the diagram in *Figure 4H* summarizes this circuit.

### Descending projection to the MdD

By far the most abundant collateral projection of vibrissa-responsive interpolaris cells is to the MdD nucleus and the dorsal horn of the cervical cord. The finding of vibrissa-responsive cells in the MdD nucleus is unexpected, as most studies reported that MdD neurons best respond to noxious stimuli (reviewed in *Martins and Tavares, 2017*; *Figure 3A–C*). Although there is no reason to believe that vibrissa deflection per se is painful, an unexpected air puff directed toward the head of a rat elicits fear-related behavior such as a startle response (*Engelmann et al., 1996*), 22 kHz ultrasonic vocalization (*Knapp and Pohorecky, 1995*), and avoidance behavior (*Cimadevilla et al., 2001*). As the MdD nucleus sends profuse projections to the cervical cord and the facial nucleus (*Bernard et al., 1990*; *Takatoh et al., 2013*; *Takatoh et al., 2021*), these results suggest that vibrissa messages to the MdD nucleus may elicit avoidance or flight behaviors. This conclusion is in line with that of the recent study, which reported that the MdD nucleus is part of a brainstem-spinal circuit to control escape responses to noxious stimuli (*Barik et al., 2018*).

The brainstem medullary reticular formation comprises two main divisions: a ventral part (MdV) and a dorsal part (MdD). The MDV nucleus stands out as specifically targeting subpopulations of forelimb-innervating motor neurons. Selective ablation or silencing experiments reveal that the MdV

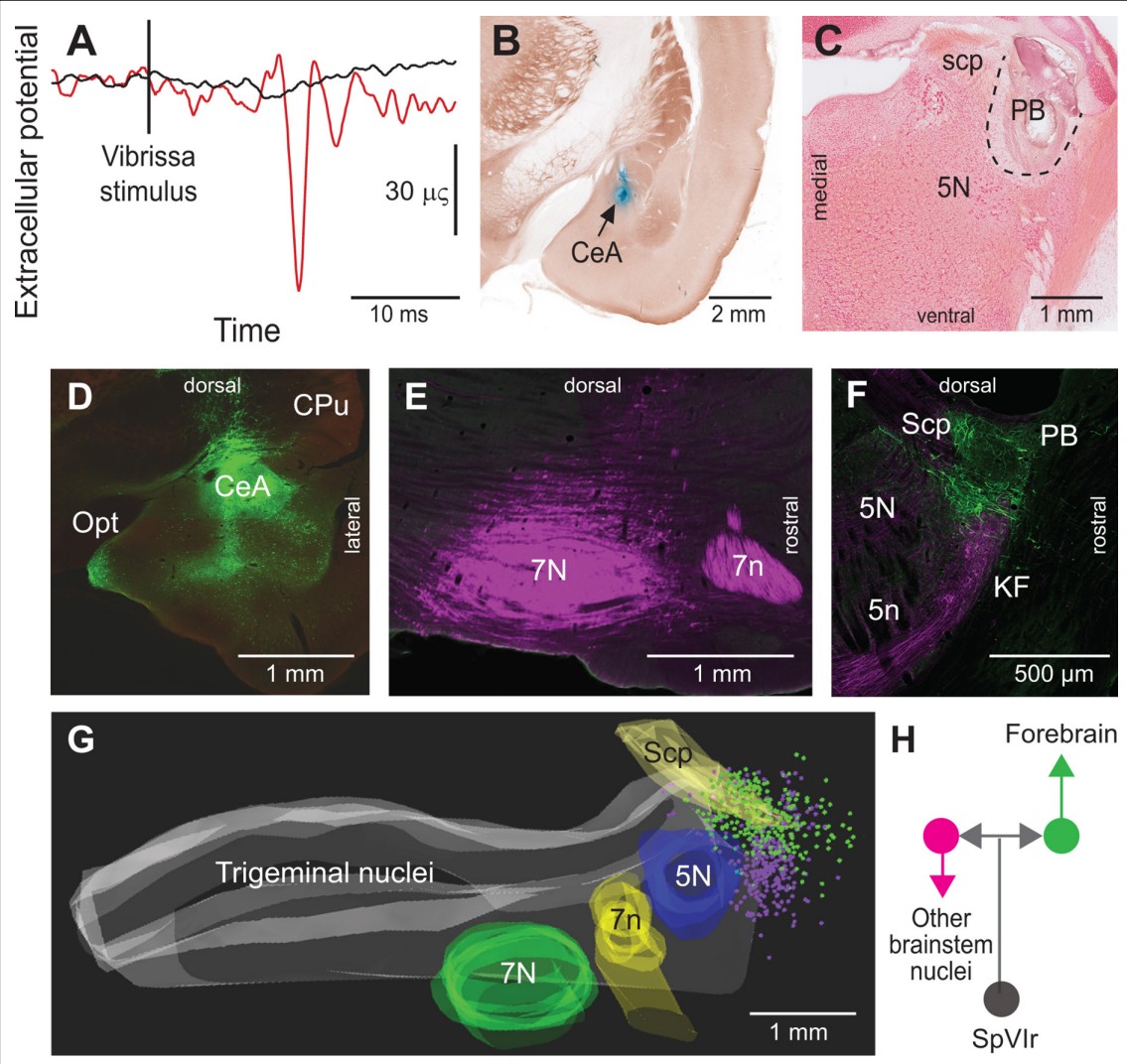

**Figure 3.** Vibrissa-evoked responses in central amygdala and anatomical evidence that separate cellular populations in the KF/PBc project to the CeA as compared to the facial nucleus. See *Figure 1* for abbreviations. (**A**) Average response (50 trials) evoked in the CeA by air puff stimulation of the vibrissae before (red trace) and after (black trace) an electrolytic lesion of the PB complex. (**B**) Recording site in the amygdala labeled by an iontophoretic injection of Chicago Sky Blue (cytochrome oxidase counterstaining). Coronal section. (**C**) Electrolytic lesion of the PB complex. Coronal section. (**D**) Injection site of retroAAV-CAG-eGFP in the CeA. Coronal section. (**E**) Injection site of retroAAV-CAG-mCherry in the facial nucleus. Sagittal section. (**F**) Retrogradely labeled cells in the KF/PBc. Sagittal section. (**G**) Sagittal view of a three-dimensional reconstruction showing the distribution of amygdala-projecting cells (green dots) in the medial and lateral PB, and facial-projecting neurons in the KF (magenta dots). Note that a few cells in lateral PB project to the facial nucleus; yet, none of the KF/PBc cells are doubly labeled. (**H**) Wiring diagram of the projections of KF and PB cells that receive vibrissa input from the interpolaris nucleus (see also Figure S3 for additional evidence).

The online version of this article includes the following figure supplement(s) for figure 3:

**Figure supplement 1.** Labeling of KF cells that project to lower brainstem (Supplementary Information related to *Figure 3*).

nucleus is critically important for skilled motor behaviors (*Esposito et al., 2014*). On the other hand, the MdD nucleus receives parabrachial inputs whose activation induces flight and escape behavior. Yet 'Activation of PB$^{(tac1)}$ neurons does not produce nocifensive responses on its own but has a major effect on responses to sensory cues. It thus seems possible that peripheral inputs provide the context necessary to trigger a specific behavioral response'. (*Barik et al., 2018*). We propose that the relay of sensory signals in the paralemniscal pathway is normally gated, and that release from gating occurs in behavioral contexts that are alarming or threatening. A recent paper (*Sun et al., 2020*) lends support for this proposal by showing that GABAergic KF/PBc cells gate the relay of aversive stimuli to limbic forebrain regions.

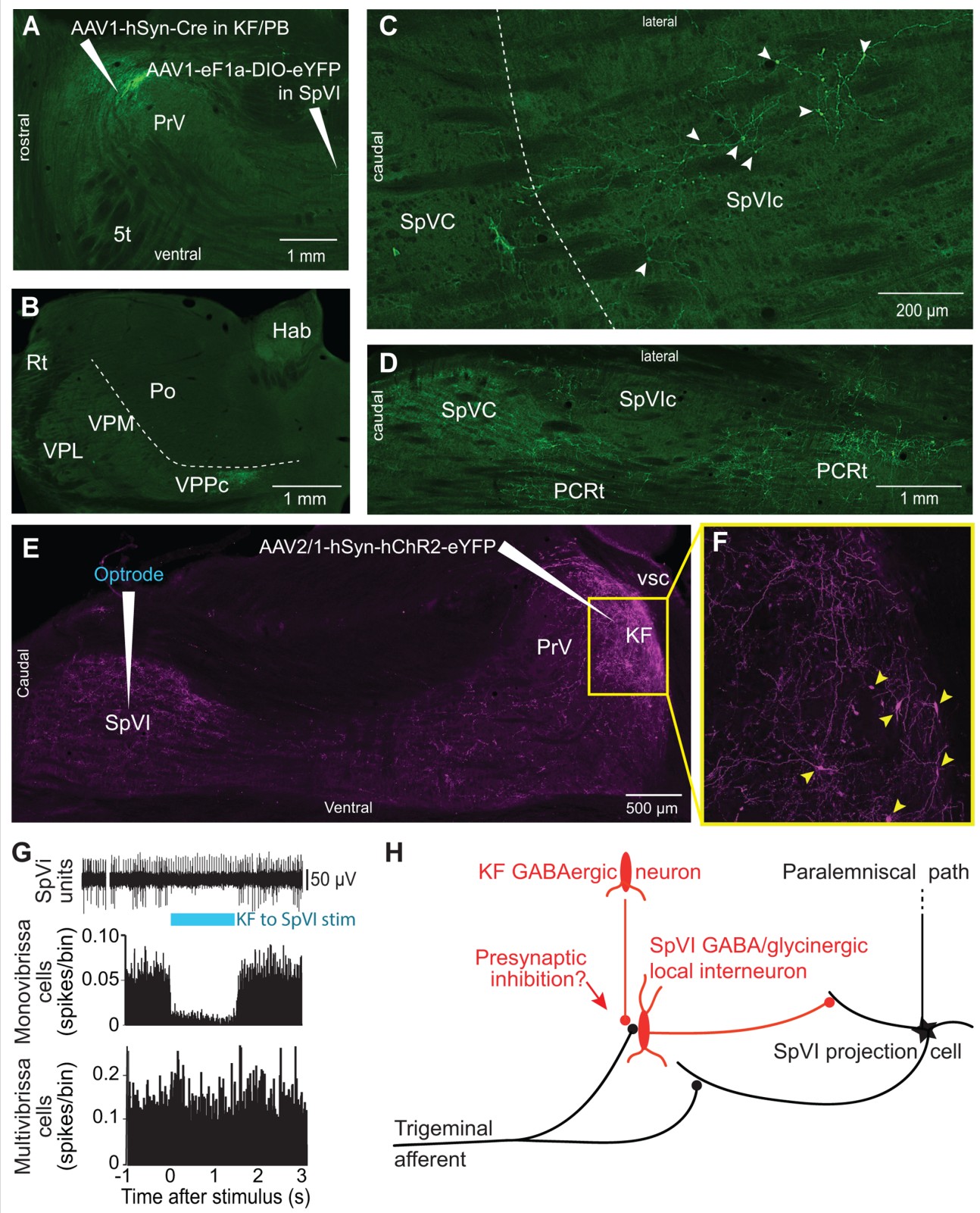

**Figure 4.** Postsynaptic targets of KF projections to the SPVI. See *Figure 1* for abbreviations. (**A**) The viral method used for transsynaptic labeling of brainstem cells that receive input from the KF. Sagittal section. (**B–D**) Transsynaptic labeling in SpVI is restricted to a small population of cells (**C**) that project to VPCC (**B**). Yet, numerous cells are transsynaptically labeled in the parvicellular reticular formation (**D**). Panel (**B**) in a coronal section while (**C**) and (**D**) are horizontal sections. (**E, F**) Viral labeling method for optogenetic stimulation of KF axons. Injection of AAV2/1-hSyn-hChR2-eYFP in KF

*Figure 4 continued on next page*

*Figure 4 continued*

results in anterograde labeling in SpVI. The framed region in (**E**) is shown in (**F**); arrowheads point to labeled cell bodies. Note the dense network of axon collaterals within the KF/PBc. Sagittal section. (**G**) Representative responses of a monovibrissa-responsive cell (biphasic unit) and a multivibrissa-responsive cell (positive unit) upon optogenetic stimulation of KF axons labeled as in (**E**). Population peristimulus time histogram shows the responses of 26 monovibrissa-responsive units upon optogenetic stimulation of KF axons along with 10 multivibrissa-responsive units to optogenetic stimulation of KF axons. Only mono-vibrissa and not multi-vibrissa cells are responsive to KF input. (**H**) Summary of the effect of KF projections on the flow of sensory input through SpVI.

## The trigemino-parabrachial amygdaloid pathway

A number of studies have reported trigeminal projections to the KF and PB nuclei (*Cechetto et al., 1985*; *Slugg and Light, 1994*; *Feil and Herbert, 1995*; *Dallel et al., 2004*; *Rodriguez et al., 2017*; see also review by *Chamberlin, 2004*). Most of these studies focused on ascending projections from subnucleus caudalis, which convey pruriceptive and nociceptive inputs to the KF/PBc (*Jansen and Giesler, 2015*). Thus, it came as a surprise to find vibrissa-responsive neurons in KF/PBc, and further find that vibrissa signals are conveyed to the limbic regions of the forebrain via the medial and lateral PB. One study previously reported short-latency activation of amygdala neurons, at about 11 ms, upon electrical stimulation of the mystacial pad (*Bernard et al., 1992*). Such a short latency is consistent with the activation of a trigemino-parabrachial-amygdaloid pathway by Aβ primary vibrissa afferents.

## The paralemniscal pathway and threat

Although vibrissa-responsive interpolaris cells do not respond to noxious stimuli, many of the regions they innervate contain cells that process nociceptive or aversive inputs. This is the case for the MdD nucleus (reviewed in *Martins and Tavares, 2017*), the KF/PBc (*Jansen and Giesler, 2015*), the superior colliculus (*Dean et al., 1989*; *Redgrave et al., 1996*), the zona incerta (*Masri et al., 2009*), and Po thalamus (*Masri et al., 2009*; *Frangeul et al., 2014*; *Sobolewski et al., 2015*). These anatomical data suggest that the paralemniscal pathway conveys signals from threatening or alarming sensory inputs, perhaps through association with aversive contexts.

In light of the evidence that interpolaris cells innervate brain regions that process nociceptive or aversive stimuli, we postulate that the KF nucleus is primarily involved in orchestrating defensive reactions to aversive and threatening stimuli. Three additional lines of evidence support this hypothesis. First, glutamatergic KF cells project to the cardio-respiratory medullary centers, the facial, hypoglossal, and ambiguous motor nuclei, the nucleus of the solitary tract, and to the preganglionic neurons of the sympathetic nervous systems (reviewed in *Saper and Stornetta, 2015*). Second, activation of KF cells elicits changes in respiration, heart rate, blood pressure (*Chamberlin and Saper, 1992*; *Lara et al., 1994*; *Guo et al., 2002*; *Chamberlin, 2004*). Third, individual KF neurons project to many of the above-mentioned targets by means of branching axons, which might form the anatomical substrate for coordinating autonomic and somatic reactions (*Song et al., 2012*).

## Sensory gating in the paralemniscal pathway

A prior study in the alert head-restrained rat reported that units in trigeminal nucleus principalis fire more spikes when the animal is whisking as opposed to not whisking (*Moore et al., 2015*). Yet the same study shows that spike rates are not significantly different between whisking and not whisking in subnucleus SpVI. This is an unexpected result, as primary vibrissa afferents discharge during whisking (*Severson et al., 2017*) and SpVI projection cells are typically driven by many more vibrissae than principalis neurons (*Furuta et al., 2006*). The observation that whisking tends to reorganize spike times of neurons in interpolaris, rather than increase overall spike rates, supports the notion that the relay of vibrissa messages in the paralemniscal pathway is normally gated as part of fast, inhibitory feedback. The inhibition of the paralemniscal pathway by KF GABAergic neurons, together with the lack of a direct connection from KF neurons to interpolaris projections cells, suggests that proposed gating occurs by a mechanism of presynaptic or extrasynaptic inhibition (*Figure 4H*).

What drives the KF nucleus, consistent with the postulated involvement in orchestrating defensive reactions to aversive and threatening stimuli? The KF nucleus receives projections from different subdivisions of the periaqueductal gray, themselves downstream of inhibitory input from the amygdala (*Tovote et al., 2016*). The release of the inhibition from the amygdala to the periaqueductal gray

may well lead to a defensive postures. It is of interest the release of inhibition provides a release from pain (*Benarroch, 2012*), also consistent with presynaptic inhibition of the trigeminus.

## Conclusion

Our results clearly indicate that the paralemniscal pathway broadcasts vibrissa-based sensory signals to a number of brainstem and forebrain regions that are involved in the expression of emotional reactions. They provide a different viewpoint on the role of the Kölliker-Fuse nucleus, which is currently considered to control the mechanics of respiratory and cardiac rhythms, but no more. We propose that the Kölliker-Fuse nucleus is involved in the larger role of orchestrating the expression of emotional reactions, for example, facial expressions, autonomic, and somatic reactions, of which respiration is one of many players.

## Materials and methods

### Subjects

Experiments were carried out in Long Evans rats of both sexes and included five juvenile rats (P30–P35) and 36 adult rats (P80–P100; 250–350 g in mass). All experiments were carried out according to the National Institutes of Health Guidelines. All experiments were approved by the Institutional Animal Care and Use Committees at Laval University and at the University of California at San Diego.

### Viruses

G-pseudotyped Lenti-Cre virus was designed and produced as described (*Nelson et al., 2013*). AAV2/1-EF1a-DIO-hChR2-EYFP-WPRE-HGHpA, AAV2/1-hSyn-Cre WPRE, AAV2/5-hSyn-eGFP-WPRE, retroAAV-CAG-tdTomato, retroAAV-CAG-eGFP, and AAV2/8-hSyn-DIO-eGFP were obtained from Addgene.

### Surgery and virus injections

All but three rats were anesthetized with ketamine (75 mg/kg) and xylazine (5 mg/kg), and body temperature was maintained at 37°C with a thermostatically controlled heating pad. Three rats were anesthetized with urethane (1.4 g/kg) to record the local field potential evoked in the amygdala by vibrissa deflection. All virus injections were carried out after electrophysiological identification of the target regions. When AAV injections were made in the interpolaris nucleus, we targeted the lateral most sector of the nucleus based on the well-known somatotopic representation of the vibrissae in this nucleus. This precaution proved useful to avoid infecting cells in neighboring regions. After 3–4 weeks of survival, based on the results of *Stanek et al., 2016*, animals were either perfused with saline and paraformaldehyde (4% (w/v) in PBS), or anesthetized with ketamine (75 mg/kg) and xylazine (5 mg/kg) for optogenetic stimulation.

In three rats, the vibrissa-responsive sector of the KF/PBc was first located by electrophysiological recording and then lesioned by passing 0.5 mA through a stainless steel electrode with a tip (10 μm) de-insulated over a length of 500 μm.

### Electrophysiology

Single-unit recordings were carried out in the KF/PBc and MdD with micropipettes (tip diameter, 1 μm) filled with either 0.5 M potassium acetate and 2% (w/v) neurobiotin, or 0.5 M potassium acetate and 4% (w/v) Chicago Sky Blue. To maximize the chance of cell recovery after juxtacellular labeling in the KF/PBc, we injected one cell per animal and perfused the rat thereafter.

A hand-held probe was first used to identify the vibrissal receptive field of MdD and KF/PBc neurons. Then we used an air jet (1.5 s duration) to deflect the vibrissae in the rostro-caudal direction. Air jets were generated by a Picospitzer (General Valve, Brookshire, TX) connected to a pipette (tip diameter, 500 μm). The pipette was positioned at a distance of 4–5 cm of the vibrissae, in front of the animal, and the air jet was directed away from the rat's face to avoid as much as possible stimulating other orofacial afferents. This created a cone-shaped air displacement that deflected six to seven whiskers. The delay between the command voltage and the actual motion of the vibrissae was measured by placing a piezoelectric film (Measurement Specialties, Fairfield, NJ) at the same distance

from the tip of the pipette (*Kleinfeld et al., 2002*). This delay was subtracted from the recordings to build PSTHs of sensory-evoked responses (20 responses; bin width, 1 ms).

Respiration was monitored with a cantilevered piezoelectric film (LDT1 028K; Measurement Specialties) resting on the rat's abdomen just caudal to the torso. All signals were sampled at 10 kHz and logged on a computer using the Labchart acquisition system (AD Instruments).

### Optogenetics

Three weeks after injection of AAV2/1-hSyn-hChR2-eYFP in the KF/PBc (three rats), we used a carbon-tip optrode (Kation Scientific) to simultaneously stimulate labeled axons and record interpolaris cells. As a test for KF-induced inhibition, we drove background spiking by jiggling a single vibrissa with a piezoelectric stimulator.

### Histology

Following perfusion brains were postfixed for 1 hr, and cryoprotected overnight in 30% (w/w) sucrose in PBS. Brains were cut at thickness of 50 μm on a freezing microtome. Labeled material was processed for either fluorescence or brightfield microscopy. For fluorescence microscopy, sections were immunoreacted with a chicken anti-GFP antibody (1:1000; Abcam), and a donkey anti-chicken Alexa 488 IgG (1:500; Jackson ImmunoResearch Labs). NeuN immunostaining was carried out with a mouse anti-NeuN antibody (1:1000; MilliporeSigma) and a goat anti-rabbit IgG conjugated to Alexa 594 (1:500; Abcam). For brightfield microscopy, sections were first counterstained for cytochrome oxidase (MilliporeSigma), and then immunoreacted with a rabbit anti-GFP antibody (1:1000; Novus Biological), a biotinylated horse anti-rabbit IgG (1:200; Vector Labs), the avidin/biotin complex (Vectastain ABC Kit; Vector Labs), and the SG peroxidase substrate (ImmPACT SG Substrate; Vector Labs). In three rats, brainstem sections were first immunoreacted with a goat anti-choline acetyltransferase antibody (1:1000; MilliporeSigma) and a rabbit anti-goat IgG conjugated to horseradish peroxidase (1:500; Abcam), which was revealed with diaminobenzidine (brown reaction product). Sections were then immunoreacted with a mouse anti-GFP antibody (1:2000; Abcam), a biotinylated goat anti-mouse IgG (1:200; Vector Labs), which was revealed with streptavidin horseradish peroxidase conjugate (Vector Labs) and the Ni-DAB substrate (black reaction product) (ImmPACT DAB Substrate; Vector Labs). Finally, the extent of electrolytic lesions was assessed on material stained with Neutral Red. Sections were scanned at a resolution of 0.5 μm/pixel (TissueScope LE; Huron Digital Pathology) and imported in Fiji or Photoshop for color and contrast adjustments.

### Data analysis

Three-dimensional maps of retrogradely labeled KF/PBc cells were constructed with the Neurolucida software (Microbrightfield).

Peristimulus time histograms were built using the LabChart 8.0 spike histogram module (AD Instruments), and the Matlab Chronux toolbox (http://www.chronux.org) was used to compute the spectral coherence between respiration and the firing rate of KF/PBc cells.

### Data

'Data sets (anatomical images and physiological time series) supporting this study, and an associated "read me" file, are deposited at Dryad, doi:10.6076/D15C7X.'.

## Acknowledgements

The authors thank Fan Wang for critical discussions and for supplying the retrograde-Lenti-Cre virus. This study was supported by grants from the Canadian Institutes of Health Research (Grant MT-5877) and the National Institutes of Health (U19 NS107466 and R35 NS097265).

## Additional information

### Funding

| Funder | Grant reference number | Author |
|---|---|---|
| National Institutes of Health | NS107466 | Martin Deschenes David Kleinfeld |
| National Institutes of Health | NS097265 | David Kleinfeld Martin Deschenes |
| Canadian Institutes of Health Research | MT-5877 | Martin Deschenes |

The funders had no role in study design, data collection and interpretation, or the decision to submit the work for publication.

### Author contributions

Michaël Elbaz, Amalia Callado Perez, Investigation; Maxime Demers, Shengli Zhao, Conrad Foo, Methodology; David Kleinfeld, Funding acquisition, Investigation, Methodology, Project administration, Resources, Supervision, Validation, Visualization, Writing - original draft, Writing - review and editing; Martin Deschenes, Data curation, Formal analysis, Funding acquisition, Investigation, Methodology, Project administration, Resources, Supervision, Validation, Visualization, Writing - original draft, Writing - review and editing

### Author ORCIDs

Michaël Elbaz http://orcid.org/0000-0002-6223-7359
David Kleinfeld http://orcid.org/0000-0001-9797-4722
Martin Deschenes http://orcid.org/0000-0002-5385-6633

### Ethics

All experiments were carried out according to the National Institutes of Health Guidelines. All experiments were approved by the Institutional Animal Care and Use Committees at Laval University (protocol number: VRR-18-068) and at the University of California at San Diego (protocol numbers S02173M and S03174).

### Decision letter and Author response

Decision letter https://doi.org/10.7554/eLife.72096.sa1
Author response https://doi.org/10.7554/eLife.72096.sa2

## Additional files

### Supplementary files

• Transparent reporting form

### Data availability

Datasets (anatomical images and physiological time series) supporting this study, and an associated "read me" file, are deposited at Dryad, https://doi.org/10.6076/D15C7X.

The following dataset was generated:

| Author(s) | Year | Dataset title | Dataset URL | Database and Identifier |
|---|---|---|---|---|
| Kleinfeld D, Deschenes M | 2021 | High resolution images and compilation of raw data | http://dx.doi.org/10.6076/dryad.D15C7X | Dryad Digital Repository, 10.5061/dryad.D15C7X |

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

# Appendix 1

## Appendix 1—key resources table

| Reagent type (species) or resource | Designation | Source or reference | Identifiers | Additional information |
|---|---|---|---|---|
| Antibody | Anti-GFP (chicken polyclonal) | Abcam | Cat#: ab13970 | 1:1000 |
| Antibody | Anti-Chicken IgY (IgG) (H+L) (donkey polyclonal) (Alexa Fluor 488) | Jackson ImmunoResearch Labs | Cat#: 703-545-155; RRID: AB_2340375 | 1:500 |
| Antibody | Anti-NeuN (mouse monoclonal (A60)) | MilliporeSigma | Cat#: MAB377 | 1:1000 |
| Antibody | Anti-Rabbit IgG – H&L (Goat polyclonal) (Alexa Fluor 594) | Abcam | Cat#: ab150080 | 1:500 |
| Antibody | Anti-GFP (rabbit polyclonal) | Novus Biologicals | Cat#: NB600-308 | 1:1000 |
| Antibody | Anti-rabbit IgG (H+L) (horse) (Biotynilated) | Vector Laboratories | Cat#: BA-1100 | 1:200 |
| Antibody | Anti-Choline-acetyltransferase (Goat polyclonal) | MilliporeSigma | Cat#: SAB2500236 | 1:1000 |
| Antibody | Anti-Goat IgG H&L (Rabbit polyclonal) (Horseradish peroxidase) | Abcam | Cat#: ab6741 | 1:500 |
| Antibody | Anti-GFP (mouse monoclonal) | Abcam | Cat#: ab1218 | 1:2000 |
| Antibody | Anti-mouse IgG (H+L) (Goat) (Biotynilated) | Vector Laboratories | Cat#: BA-9200 | 1:200 |
| Recombinant DNA reagent | G-pseudotyped-Lenti-Cre | Fan Wang (MIT McGovern Institute) | | |
| Recombinant DNA reagent | pAAV-hSyn-DIO-eGFP (AAV8) | Addgene | Addgene ID: 50457 | Depositor: Bryan Roth |
| Recombinant DNA reagent | pAAV-Ef1a-DIO-hChR2-eYFP-WPRE-HGHpA (AAV1) | Addgene | Addgene ID: 20298 | Depositor: Karl Deisseroth. New version of this virus is pAAV-EF1a-double floxed-hChR2(H134R)-EYFP-WPRE-HGHpA |
| Recombinant DNA reagent | pENN.AAV.hSyn.Cre.WPRE.hGH (AAV1) | Addgene | Addgene ID: 105553 UPenn ID: AV-1-PV-2676 | Depositor: James M. Wilson |
| Recombinant DNA reagent | pAAV.hSyn.eGFP.WPRE.bGH (AAV5) | Addgene | Addgene ID: 105539 UPenn ID: AV-5-PV-1696 | Viral service discontinued |
| Recombinant DNA reagent | pAAV-CAG-tdTomato (AAV retrograde) | Addgene | Addgene ID: 59462 | Depositor: Edward Boyden |
| Recombinant DNA reagent | pAAV-CAG-eGFP (AAV retrograde) | Addgene | Addgene ID: 37825 | Depositor: Edward Boyden |
| Peptide, recombinant protein | Horseradish Peroxidase Streptavidin (HRP-Streptavidin conjugate) | Vector Laboratories | Cat#: SA-5704-100 | |
| Commercial assay or kit | Vectastain Elite ABC-HRP Kit, Peroxidase (Standard) | Vector Laboratories | Cat#: PK-6100 | |
| Commercial assay or kit | ImmPACT SG Substrate, Peroxidase (HRP) | Vector Laboratories | Cat#: SK-4705 | |

*Appendix 1 Continued on next page*

*Appendix 1 Continued*

| Reagent type (species) or resource | Designation | Source or reference | Identifiers | Additional information |
|---|---|---|---|---|
| Commercial assay or kit | ImmPACT DAB Substrate, Peroxidase (HRP) | Vector Laboratories | Cat#: SK-4105 | |
| Chemical compound, drug | Chicago Sky Blue 6B | MilliporeSigma | Cat#: C8679 | Powder |
| Chemical compound, drug | Neutral Red | MilliporeSigma | Cat#: 861251 | |
| Software, algorithm | Neurolucida | MBF Bioscience | RRID:SCR_001775 | http://mbfbioscience.com/neurolucida |
| Software, algorithm | LabChart 8.0 Spike Histogram Module | AD Instruments Data Acquisition Systems for Life Science | RRID:SCR_001620 | https://www.adinstruments.com/ |
| Software, algorithm | Matlab Chronux Toolbox | Chronux (MATLAB) | RRID:SCR_005547 | http://chronux.org |
| Other | Neurobiotin Tracer | Vector Laboratories | Cat#: SP-1120 | |
| Other | Cytochrome c oxidase (from bovine heart) | MilliporeSigma | Cat#: C5499 | |

