## [Editor Report]

Elbaz and colleagues show that the interpolaris subdivision of the trigeminal brainstem innervates not only somatosensory thalamus but also other midbrain and hindbrain structures. A key target of interpolaris appears anatomically to be excitatory projection neurons of the Kolliker-Fuse nucleus. Physiological recordings from the dorsal medullary reticular nucleus, Kolliker-Fuse nucleus, and the central amygdala reveal responses to vibrissa deflection, suggesting a role for this pathway in limbic system behaviors.

---

## [Decision Letter]

**Decision letter after peer review:**

Thank you for submitting your article "A vibrissa pathway that activates the limbic system" for consideration by *eLife*. Your article has been reviewed by 3 peer reviewers, one of whom is a member of our Board of Reviewing Editors, and the evaluation has been overseen by John Huguenard as the Senior Editor. The reviewers have opted to remain anonymous.

Essential revisions:

1) Please increase clarity over some details of the G-pseudotyped Lenti-cre virus

Can the authors expand on aspects of this G-pseudotyped Lenti-cre virus? e.g, in Figure 1, are the authors certain that only monosynaptic projections are labelled? Is there a possibility of secondary jumping? If there is a possibility of secondary jumping, how did the authors choose the survival time of the rats?

Relatedly, Stanek is referenced several times, but there's no entry in the bibliography. If it's the JNeurosci paper from Fan Wang, those authors did not use the term "G-pseudotyped", so I think the authors here need to fully explain what this means and how the virus works.

2) Paper framing and setup of the KF nucleus

While the starting point of this paper is about one pathway, the paper also includes substantial data related to an entirely different pathway and topic: namely, interpolaris and the Kolliker-Fuse nucleus. The background of these nuclei are only hinted at and only at the end. More to the topic, the authors do discuss a whisker-limbic pathway, which is interesting, but their data also suggest additional functions of these pathways other than avoidance and flight behaviors, which could be discussed. Similarly, the authors zero in on their results regarding KF GABA neurons, but never discuss what might even conceivably be driving these cells. Some rounding out of the concepts, both known and hypothetical, would really improve this paper. Separately, this is the first paper to ever study the physiological responses of several of these structures to whisker stimulation, but the manuscript says absolutely nothing about the nature of these stimuli (see point 3 below).

(2a) Intro, 1st paragraph, "…two main trigeminothalamic pathways". While there are anatomical connections between these structures, it is unclear the degree to which the paralemniscal connections serve as a main route for ascending signals and may have other functions. The idea of two parallel pathways is largely speculative and not based on functional studies. The caveat should be noted rather than presented as a consensus or textbook view. It is also not the focus of this study and somewhat distracting.

(2b) The second to last sentence of the Conclusion reflects one of the biggest take home messages of this paper: its implications for the function(s) of the Kolliker-Fuse nucleus. From my point of view, this is also a paper about SpVI-KF paper. I would encourage the authors to bring up the SpVI/KF nuclei earlier in the manuscript to explain their background to the naïve reader who is unlikely to be familiar with them and then in the Discussion better explain the implications.

(2c) pp.7-8 focus on a role of MdD and cervical cord innervation in avoidance or flight behaviors, which seems overly narrow. If the authors are serious about this suggestion, they should explain how this would work. Do the authors really think that certain kinds of whisker stimuli but not others could trigger flight? Which ones? This seems odd and not terribly ethological. Perhaps a more ethological scenario is that some of these subcortical targets are used for shaping on-going movements in response to whisker stimuli rather than triggering whole new behaviors. For instance, subcortical pathways are sufficient for mice to shape their limb movements as they jump over obstacles they detect with their whiskers (Warren and Sawtell 2021 *eLife*). A Po-MdD or Po-cervical cord pathway would be prime candidates for that kind of sensory-motor behavior and many others.

(3) Please increase clarity around sensory stimuli and responses

(3a) The authors mention on line 5 of page 5 that 'the receptive field of KF/PBc and MdD cells include multiple vibrissae,' but there is no mention of the receptive field size, or a panel showing how responses of KF/PBc and MdD cells change with stimulation of different whiskers. Could the authors clarify this point?

(3b) One of the nice aspects of this paper is that the experimenters physiologically checked for whisker-evoked responses in structures not previously examined by the field. However, there are absolutely no details about the parameters of the whisker stimulus whatsoever: the nature of movement (square? sine? sawtooth? ramp-and-hold?), the amplitude, the velocity, the duration. If repetitive, what is the frequency? On a related note, pp.7,22, "jiggling" is a colloquialism and not a standard word for describing the manipulation of the whiskers. "deflecting", "stimulating"? Did the authors use different sets of sensory stimuli (e.g., changes in magnitude, direction, or frequency of stimulus, etc.) while recording KF/PBc or MdD neurons to see whether KF/PBc or MdD responses in the same whisker change with parameters of the stimulus?*Reviewer #1:*

This work presents an advance over previous studies by providing anatomical evidence for a potential role of the paralemniscal pathway in responses to threat. High quality anatomical studies in these deep brainstem pathways are particularly important for identifying circuit architectures for high stakes sensorimotor processing, such as response to immediate threat.

Below are some issues that, if addressed, would strengthen this study and make it more suitable for publication:

(1) Could the authors provide motivation the use of juvenile rats for the tracing experiments in Figure 1, but adult rats for all other figures provided?

A previous study (Stanek et al., 2014 – *eLife*) performed similar circuit tracing experiments investigating the diversity of masseter and genioglossus premotor neurons in mice, and provided evidence for transient connections between DCN and XII motor neurons in P1 – P8, but not P8 – P15, age mice. Although Figure 1 – supplemental figure 1 suggests there may not be such differences between age groups, did the authors find any differences in connectivity between juvenile and adult rats?

(2) Can the authors expand on aspects of this G-pseudotyped Lenti-cre virus? e.g, in Figure 1, are the authors certain only monosynaptic projections are labelled? Is there a possibility of secondary jumping? If there is a possibility of secondary jumping, how did the authors choose the survival time of the rats?

(4) The authors mention on line 5 of page 5 that 'the receptive field of KF/PBc and MdD cells include multiple vibrissae,' but there is no mention of the receptive field size, or a panel showing how responses of KF/PBc and MdD cells change with stimulation of different whiskers. Could the authors clarify this point?

(5) Did the authors use different sets of sensory stimuli (e.g., changes in magnitude, direction, or frequency of stimulus, etc.) while recording KF/PBc or MdD neurons to see whether KF/PBc or MdD responses in the same whisker change with parameters of the stimulus?

The authors provide citations and propose that KF/PBc and MdD are involved in processing threatening or noxious stimuli, and projections from interpolaris cells to these regions may be involved in the expression of emotional reactions. The authors also provide evidence that unexpected air puffs directed towards the head of rats elicits fear-related behavior, which may be driven via the vibrassal system.

If interpolaris projections to KF/PBc or MdD are involved in the processing of noxious stimuli, how exactly is a valence, such as threatening or non-threatening, assigned to a particular whisker stimulus, but not others? This may be done through multiple mechanisms, but one possibility is that the magnitude, direction, or frequency of a stimulus may elicit different responses. For example, a rat may respond very differently to an airpuff that acts at a single whisker versus a gust of wind that acts on all.

(6) The projections from interpolaris cells to MdD are interesting. A recent study (Ruder et al., 2021 – Nature) found that optogenetic activation of MdD-projecting, but not MdV- or spinal cord-projecting, lateral rostral medulla (LatRM) neurons elicited grooming sequences or hand-to-mouth movements in mice. Although the authors of the current study focus on the role of MdD in fear and threat responses in their Discussion, one possibility is that the projections from SpVI to MdD may provide a short latency pathway to elicit a grooming sequence in response to potentially aversive stimuli on the whisker pad (e.g., dirt/mud on the whiskers). The authors may want to add a few sentences in their discussion regarding this point.

(7) While the authors' data suggest sensory gating in the paralemniscal pathway is by presynaptic inhibition, and provide further citations in the Discussion to strengthen this point, there is no discussion as to what this may mean for the animal's behavior or the computations performed in this circuit. Could the authors expand in the Discussion on what gating of the paralemniscal pathway might suggest? In other words, why might this pathway need to be gated, and what does this mean with respect to behavior (for example, the response to noxious/threatening stimuli)?

While the authors' findings suggest gating of the paralemniscal pathway is via presynaptic inhibition, could the authors expand on why gating via presynaptic inhibition from KF to interpolaris cells might be computationally advantageous in this circuit, relative to other mechanisms such as postsynaptic inhibition?*Reviewer #2:*

This Short Report from Elbaz and coauthors investigated the connectivity of one of the vibrissal brainstem nuclei (interpolaris, SpVI), thought by some to belong to a "paralemniscal" pathway from the sensory periphery to the cerebral cortex. Using sophisticated viral tracing methods, they show that individual interpolaris cells can diverge to target multiple midbrain and hindbrain structures, including MdD and the cervical spinal cord. They then show that Kolliker-Fuse neurons receiving interpolaris inputs project to numerous targets but do not feedback onto interpolaris. Physiological recordings from MdD, KF, and the central amygdala reveal responses to vibrissa deflection. Using optogenetics, they show that KF GABAergic interneurons, however, do project back to interpolaris and may gate sensory responses. They conclude that this circuitry may allow the whisker system to engage the limbic system.

The viral tracing methods are state-of-the-art, and the authors put them to good use not only confirming known interareal connections but also revealing novel ones. The physiology data are nice complements to the anatomical experiments, demonstrating that these connections can relay sensory signals and that several targets are vibrissa responsive. They also suggest that the KF GABA neurons may participate in a feedback inhibitory circuit or descending inhibitory circuit, rather than a feedforward one.

The main weaknesses of the paper are largely related to interpretation and presentation, which often leaves the reader confused. While the starting point of this paper is about one pathway, the paper is really about an entirely different pathway and topic: namely, interpolaris and the Kolliker-Fuse nucleus. The background of these nuclei are only hinted at and only at the end. More to the topic, the authors do discuss a whisker-limbic pathway, which is interesting, but their data also suggest additional functions of these pathways other than avoidance and flight behaviors, which could be discussed. Similarly, the authors zero in on their results regarding KF GABA neurons, but never discuss what might even conceivably be driving these cells. Some rounding out of the concepts, both known and hypothetical, would really improve this paper. Separately, this is the first paper to ever study the physiological responses of several of these structures to whisker stimulation, but the manuscript says absolutely nothing about the nature of these stimuli.

1. Intro, 1st paragraph, "…two main trigeminothalamic pathways". While there are anatomical connections between these structures, it is unclear the degree to which the paralemniscal connections serve as a main route for ascending signals and may have other functions. The idea of two parallel pathways is largely speculative and not based on functional studies. The caveat should be noted rather than presented as a consensus or textbook view. It is also not the focus of this study and somewhat distracting.

2. The second to last sentence of the Conclusion reflects one of the biggest take home messages of this paper: its implications for the function(s) of the Kolliker-Fuse nucleus. From my point of view, this is really an SpVI-KF paper, not a Po/paralemniscal pathway. I would encourage the authors to bring up the SpVI/KF nuclei earlier in the manuscript to explain their background to the naïve reader who is unlikely to be familiar with them and then in the Discussion better explain the implications.

3. One of the nice aspects of this paper is that the experimenters physiologically checked for whisker-evoked responses in structures not previously examined by the field. However, there are absolutely no details about the parameters of the whisker stimulus whatsoever: the nature of movement (square? sine? sawtooth? ramp-and-hold?), the amplitude, the velocity, the duration. If repetitive, what is the frequency? On a related note, pp.7,22, "jiggling" is a colloquialism and not a standard word for describing the manipulation of the whiskers. "deflecting", "stimulating"?

4. pp.7-8 focus on a role of MdD and cervical cord innervation in avoidance or flight behaviors, which seems overly narrow. If the authors are serious about this suggestion, they should explain how this would work. Do the authors really think that certain kinds of whisker stimuli but not others could trigger flight? Which ones? This seems odd and not terribly ethological. Perhaps a more ethological scenario is that some of these subcortical targets are used for shaping on-going movements in response to whisker stimuli rather than triggering whole new behaviors. For instance, subcortical pathways are sufficient for mice to shape their limb movements as they jump over obstacles they detect with their whiskers (Warren and Sawtell 2021 *eLife*). A Po-MdD or Po-cervical cord pathway would be prime candidates for that kind of sensory-motor behavior and many others.

5. Stanek is referenced several times, but there's no entry in the bibliography. If it's the JNeurosci paper from Fan Wang, those authors did not use the term "G-pseudotyped", so I think the authors here need to fully explain what this means and how the virus works.

6. What-either known or hypothetically-could activate the KF GABA neurons (e.g., shown in Figure 4H and discussed several times)? If state-dependent GABAergic gating is an idea the authors wish to put forward here, they could elaborate a bit more.

*Reviewer #3:*

The authors address the important problem of defining the functional role of the paralemniscal pathway for ascending somatosensory information. This major pathway's function remains relatively obscure, at least in comparison to the lemniscal pathway. The authors' focus is on the whisker system. Their main experimental objective therefore was to anatomically and physiologically map the collaterals of SpVi neurons that respond to whisker inputs and project to the PO thalamus. They use viral and conventional labeling methods together with electrophysiology in anesthetized rats.

Through difficult experiments, the authors show that these SpVi neurons have broad collateral projections to numerous midbrain and hindbrain targets, including to the dorsal medullary reticular nucleus (MdD) and the Kollicker-Fuse/parabrachial complex (KF/PB). They show that neurons in the MdD and KF/PB respond to whisker input at relatively short latency, implying drive from the projections from SpVi. They further trace projections emanating from the KF/PB neurons that receive input from SpVi, following the circuit downstream, and find that the areas previously known to receive KF/PB input do in fact receive such input from the SpVi-recipient neurons. Interestingly, they do not observe KF/PB feedback to the trigeminal nuclei. The authors show that the central amygdala responds to whisker input in rats, consistent with a projection from the PB and a possible role in mediating aversive responses to whisker input. The authors also find that KF/PB neurons are segregated into those projecting to forebrain and to lower brainstem targets. Finally, the authors provide physiological and anatomical evidence that KF neurons can gate sensory input flowing through the SpVi nucleus via a presynaptic or extrasynaptic mechanism.

The experiments and analyses are carefully performed and the overall quality of evidence for the authors' conclusions is strong. This is a substantial set of observations of value to the field. The authors succeed in their objective of mapping functional pathways originating from the SpVi neurons of the paralemniscal pathway. In so doing they shed light on the function of the paralemniscal pathway.

It is sometimes not clear how many rats the data shown in figures comes from, leading to a bit of a mismatch between the text, which describes findings across a population of rats, and the figures.

---

## [Author Response]

Essential revisions:1) Please increase clarity over some details of the G-pseudotyped Lenti-cre virusCan the authors expand on aspects of this G-pseudotyped Lenti-cre virus? e.g, in Figure 1, are the authors certain that only monosynaptic projections are labelled? Is there a possibility of secondary jumping? If there is a possibility of secondary jumping, how did the authors choose the survival time of the rats?

The pseudotyped retrograde lenti virus-cre was devised and supplied by our colleague and collaborator Prof. Fan Wang (MIT). It was first described in Nelson et al., (2013).

Nelson A, Schneider DM, Takatoh J, Sakurai K, Wang F, Mooney R. 2013. A circuit for motor cortical modulation of auditory cortical activity. J. Neurosci. 33:14342-14353.

Lenti virus are enveloped viruses. Viral fusion glycoproteins (FuG proteins) are essential for enveloped virus infection as these proteins mediate fusion between the virus envelope and host cellular membrane to release the viral genome into cells. The G protein is not replicated upon viral infection. Thus the virus cannot jump across synapses. As a practical issue, we did not observe cell body labeling outside the interpolaris nucleus. Survival time was based on prior studies using this virus (Stanek et al., 2016).

Stanek, E., Rodriguez, E., Zhao, S., Bao-Xia Han, B.-X., and Wang, F. (2016) Supratrigeminal bilaterally projecting neurons maintain basal tone and enable bilateral phasic activation of jaw-closing muscles. J. Neurosci. 36, 7663-7675.

To cross-check results obtained with G-pseudotyped lentivirus we injected the anterograde virus AAV1-hSyn-eGFP-WPRE in the vibrissa-responsive sector of the SpVIr (3 adult rats). We found similar results.

Relatedly, Stanek is referenced several times, but there's no entry in the bibliography. If it's the JNeurosci paper from Fan Wang, those authors did not use the term "G-pseudotyped", so I think the authors here need to fully explain what this means and how the virus works.

Thank you for noting this error. We now include Nelson et al., 2013 and Stanek et al., (2015) as references.

2) Paper framing and setup of the KF nucleusWhile the starting point of this paper is about one pathway, the paper also includes substantial data related to an entirely different pathway and topic: namely, interpolaris and the Kolliker-Fuse nucleus. The background of these nuclei are only hinted at and only at the end.

We agree with the reviewer that the studies on the Kölliker-Fuse/parabrachial complex constitute an important and novel part of our findings. We now motivate these studies in the Introduction with the text.

"with a focus on pathways that include the Kölliker-Fuse/parabrachial complex (KF/PBc). The KF/PBc were previously described primarily in terms of their role in eliciting changes in respiration and cardiac function (reviewed in Saper and Stornetta, 2015). Yet the role of inhallation in either driving or pacing whisking (Deschênes et al., 2012**;** Moore et al., 2013) provides motivation to investigate if KF/PBc has a broader role in orofacial motor actions. Further motivation comes from the projections of the KF to the trigeminal sensory nuclei (Geerling et al., 2017), which contains multiple pathways for the control of motoneurons involved in whisking (Bellavance et al., 2017). Lastly, we searched for feedback projections from targets along the paralemnical pathways to brainstem nuclei as a means to uncover a potential role of this pathway in rat’s behavior.”

More to the topic, the authors do discuss a whisker-limbic pathway, which is interesting, but their data also suggest additional functions of these pathways other than avoidance and flight behaviors, which could be discussed. Similarly, the authors zero in on their results regarding KF GABA neurons, but never discuss what might even conceivably be driving these cells.

We now add material to the Discussion relevant to autonomic "avoidance and flight behaviors" and consider projections from different subdivisions of the periaqueductal gray that themselves are downstream of inhibitory input from the amygdala (Tovote et al., 2016). If is important to note that driving KF by release of inhibition from amygdala provides release from pain as well as defensive postures (Benarroch 2012). Both possibilities are consistent with presynaptic inhibition of the trigeminus following disinhibition, i.e., turning off, to the amygdala.

The new text is

“What drives the Kölliker-Fuse, consistent with the postulated involvement in orchestrating defensive reactions to aversive and threatening stimuli? Kölliker-Fuse receives projections from different subdivisions of the periaqueductal gray, themselves downstream of inhibitory input from the amygdala (Tovote et al., 2016). The release of the inhibition from the amygdala to the periaqueductal gray may well lead to a defensive postures. It is of interest the release of inhibition provides a release from pain (Benarroch 2012), also consistent with presynaptic inhibition of the trigeminus.”

Tovote, P., Esposito, M.S., Botta, P., Chaudun, F., Fadok, J.P., Markovic, M., Wolff, S.B.E., Ramakrishnan, C., Fenno, L., Deisseroth, K., Herry, C., Arber, S., Lüthi, A. (2016) Midbrain circuits for defensive behaviour. Nature 534, 206-216.

Benarroch, E. E. (2012) Periaqueductal gray: An interface for behavioral control. Neurology 78, https://doi.org/10.1212/WNL.0b013e31823fcdee

Some rounding out of the concepts, both known and hypothetical, would really improve this paper. Separately, this is the first paper to ever study the physiological responses of several of these structures to whisker stimulation, but the manuscript says absolutely nothing about the nature of these stimuli (see point 3 below).

We have attempted to "round out the concepts" as noted above. We apologize for not including details of stimulation parameters, etc., and have added them to the Methods as noted below.

(2a) Intro, 1st paragraph, "…two main trigeminothalamic pathways". While there are anatomical connections between these structures, it is unclear the degree to which the paralemniscal connections serve as a main route for ascending signals and may have other functions. The idea of two parallel pathways is largely speculative and not based on functional studies. The caveat should be noted rather than presented as a consensus or textbook view. It is also not the focus of this study and somewhat distracting.

The idea of two parallel pathways is not based on functional studies. We state in the first paragraph of the Introduction that the anatomical evidence, based on tract tracing, is unequivocal for two pathways. We then address the unresolved issue of function, as requested by the reviewer, in chapter three of the Introduction.

That the paralemniscal pathway exerts other function(s) than coding for stimuli parameters is precisely the point of our paper, and we appreciate the reviewer's enthusiasm for this point. We originally wrote:

"While the lemniscal pathway conveys tactile information, as well as information about the relative phase of the vibrissae in the whisk cycle (Yu et al., 2006; Curtis and Kleinfeld, 2009; Khatri et al., 2010; Moore et al., 2015; Isett and Feldman, 2020), the role of the paralemniscal pathway remains puzzling. It was proposed that this pathway conveys information about whisking kinematics (Yu et al., 2006; Golomb et al., 2003), but later studies found that encoding of whisking along the paralemniscal pathway is relatively poor (Moore et al., 2015; Urbain et al., 2015). It was also proposed that the paralemniscal pathway is specifically activated upon noxious stimulation (Masri et al., 2009; Frangeul et al., 2014), but it has never been shown that interpolaris cells that respond to vibrissa deflection are also activated by noxious stimuli. Thus the general function of the paralemnical pathway remains unresolved."

We conclude by showing that the paralemniscal pathway carries both ascending and descending signals to brain regions that are involved in the expression of emotional reactions; KF/PB in particular; and the MdD; please see papers by Barik et al., (2018) and Palmiter (2018).

Barik et al., (2018) A brainstem-spinal circuit controlling nocifensive behavior. Neuron 100, 14911503.

Palmiter RD (2018) The parabrachial nucleus: CGRP neurons function as a general alarm. Trends in Neurosci. 41, 280-293.

(2b) The second to last sentence of the Conclusion reflects one of the biggest take home messages of this paper: its implications for the function(s) of the Kolliker-Fuse nucleus. From my point of view, this is also a paper about SpVI-KF paper. I would encourage the authors to bring up the SpVI/KF nuclei earlier in the manuscript to explain their background to the naïve reader who is unlikely to be familiar with them and then in the Discussion better explain the implications.

We agree and, as noted – we have added to the Introduction and Discussion as noted above.

(2c) pp.7-8 focus on a role of MdD and cervical cord innervation in avoidance or flight behaviors, which seems overly narrow. If the authors are serious about this suggestion, they should explain how this would work. Do the authors really think that certain kinds of whisker stimuli but not others could trigger flight? Which ones? This seems odd and not terribly ethological.

We expand on this point in response to the reviewer. The brainstem medullary reticular formation ventral part (MdV – not MdD) stands out as specifically targeting subpopulations of forelimbinnervating motor neuronsWe do not suggest that certain kinds of whisker stimuli trigger flight. We have added a section to the Discussion on "Descending Projection to the MdD".

The brainstem medullary reticular formation comprises two main divisions: a ventral part (MdV) and a dorsal part (MdD). The MDV stands out as specifically targeting subpopulations of forelimb-innervating motor neurons. Selective ablation or silencing experiments reveal that MdV, is critically important for skilled motor behaviors (Esposito et al., 2014). On the other hand the MdD receives parabrachial inputs whose activation induces flight and escape behavior. Yet "Activation of PB^(tac1)^ neurons does not produce nocifensive responses on its own but has a major effect on responses to sensory cues. It thus seems possible that peripheral inputs provide the context necessary to trigger a specific behavioral response." (Barik et al., 2018). We propose that the relay of sensory signals in the paralemniscal pathway is normally gated, and that release from gating occurs in behavioral contexts that are alarming or threatening. A recent paper (Sun et al., 2020) lends support for this proposal by showing that GABAergic KF/PBc cells gate the relay of aversive stimuli to limbic forebrain regions.

Barik, A., Thompson, J.H., Seltzer, M., Ghitani, N., and Chesler, A.T. (2018). A brainstem-spinal circuit controlling nocifensive behavior. Neuron 100, 1491-1503.

Esposito, M.S., Capelli, P. and Arber, S. (2014) Brainstem nucleus MdV mediates skilled forelimb motor tasks. Nature 508: 351-356.

Sun, L., Liu, R., Guo, F. Wen, W.-q., Ma, X.-l., Li, K.-y., Sun, H. Xu, C.-l., Li, Y.-y.. Wu, M.-y., Zhu, Z-g., Li, X.-j.,Yu, Y.-q., Chen, Z., Li, X.-y., and Duan, S. (2020) Parabrachial nucleus circuit governs neuropathic. Nat. Comm. 11, e5974.

Perhaps a more ethological scenario is that some of these subcortical targets are used for shaping on-going movements in response to whisker stimuli rather than triggering whole new behaviors. For instance, subcortical pathways are sufficient for mice to shape their limb movements as they jump over obstacles they detect with their whiskers (Warren and Sawtell 2021 eLife). A Po-MdD or Po-cervical cord pathway would be prime candidates for that kind of sensory-motor behavior and many others.

We thank the reviewer for her/his suggestion. Feedback to SpVI would modulate the pathway from SpVI to the facial motor nucleus that control the contact time within a whisk cycle (Sachdev et al., 2003; Deutsch et al., 2012). However, we feel that discussion of this possibility and other potentially ethological issues are outside the scope of the current work.

Sachdev, R.H.S., Berg, R.W., Chompney, G., Kleinfeld, G. and Ebner, F.F. (2003) Somatosensory and Motor Research 20:162-169.

Deutsch, D., Pietr, M., Knutsen, P.M., Ahissar, E., and Schneidman, E. (2012). Fast feedback in active sensing: Touch-induced changes to whisker-object interaction. Public Library of Science ONE *7*, e44272.

(3) Please increase clarity around sensory stimuli and responses(3a) The authors mention on line 5 of page 5 that 'the receptive field of KF/PBc and MdD cells include multiple vibrissae,' but there is no mention of the receptive field size, or a panel showing how responses of KF/PBc and MdD cells change with stimulation of different whiskers. Could the authors clarify this point?

We did not investigate the parameters of vibrissa deflection that best drive KF/PBc and MdD cells. Our goal was to cross-check the tract tracing results which indicate that some KF/PBc and MdD cells receive input from vibrissa-responsive interpolaris neurons.

(3b) One of the nice aspects of this paper is that the experimenters physiologically checked for whisker-evoked responses in structures not previously examined by the field. However, there are absolutely no details about the parameters of the whisker stimulus whatsoever: the nature of movement (square? sine? sawtooth? ramp-and-hold?), the amplitude, the velocity, the duration. If repetitive, what is the frequency?

We added this paragraph to the Methods section; the method was previously used in Kleinfeld et al., (2002).

“A hand-held probe was first used to identify the vibrissal receptive field of MdD and KF/PB neurons. Then we used an air jet (1.5 s duration) to deflect the vibrissae in the rostro-caudal direction. Air jets were generated by a Picospitzer (General Valve, Brookshire, TX) connected to a pipette (tip diameter, <inline-graphic mime-subtype="png" mimetype="image" xlink:href="media/image1.png" />500 μm). The pipette was positioned at a distance of 4 – 5 cm of the vibrissae, in front of the animal, and the air jet was directed away from the rat's face to avoid as much as possible stimulating other orofacial afferents. This created a cone-shaped air displacement that deflected six to seven whiskers. The delay between the command voltage and the actual motion of the vibrissae was measured by placing a piezoelectric film (Measurement Specialties, Fairfield, NJ) at the same distance from the tip of the pipette (Kleinfeld et a;. 2002). This delay was subtracted from the recordings to build PSTHs of sensory-evoked responses (20 responses; bin width, 1 ms).”

Kleinfeld, D., Sachdev, R.N.S., Merchant, L.M., Jarvis, M.R., and Ebner, F.F. (2002). Adaptive filtering of vibrissa input in motor cortex of rat. Neuron *34*, 1021-1034.

On a related note, pp.7,22, "jiggling" is a colloquialism and not a standard word for describing the manipulation of the whiskers. "deflecting", "stimulating"? Did the authors use different sets of sensory stimuli (e.g., changes in magnitude, direction, or frequency of stimulus, etc.) while recording KF/PBc or MdD neurons to see whether KF/PBc or MdD responses in the same whisker change with parameters of the stimulus?

We did not investigate the parameters of vibrissa deflection that best drive KF/PB and MdD cells. Here our goal was to cross-check the tract tracing results which indicate that some KF/PB and MdD cells receive input from vibrissa-responsive interpolaris neurons.